# Monocular Facial Presentation–Attack–Detection: Classifying Near-Infrared Reflectance Patterns

**Ali Hassani** [1,*], **Jon Diedrich** [2] **and Hafiz Malik** [1]

1   Information Systems, Security and Forensics Lab, University of Michigan-Dearborn,
    Dearborn, MI 48128, USA
2   Research and Advanced Engineering, Ford Motor Company, Dearborn, MI 48124, USA
*   Correspondence: alihassa@umich.edu

**Featured Application: Face-recognition for monocular systems (e.g., phones and buildings).**

**Abstract:** This paper presents a novel material spectroscopy approach to facial presentation–attack–defense (PAD). Best-in-class PAD methods typically detect artifacts in the 3D space. This paper proposes similar features can be achieved in a monocular, single-frame approach by using controlled light. A mathematical model is produced to show how live faces and their spoof counterparts have unique reflectance patterns due to geometry and albedo. A rigorous dataset is collected to evaluate this proposal: 30 diverse adults and their spoofs (paper-mask, display-replay, spandex-mask and COVID mask) under varied pose, position, and lighting for 80,000 unique frames. A panel of 13 texture classifiers are then benchmarked to verify the hypothesis. The experimental results are excellent. The material spectroscopy process enables a conventional MobileNetV3 network to achieve 0.8% average-classification-error rate, outperforming the selected state-of-the-art algorithms. This demonstrates the proposed imaging methodology generates extremely robust features.

**Keywords:** face; liveliness; PAD; monocular; texture

## 1. Introduction

Face recognition (FR) is becoming the go-to authentication technology. It is convenient: by simply looking at a camera, a person can instantly verify their identity. This is made possible through advances in deep learning [1]; algorithms can now discern a face from over 50,000 [2]. This convenient precision is yielding mass adoption, where over 100 million smart-phones alone employ FR [3]. Other industries are also quickly following suit. High-tech offices have begun to offer it as a means of building access [4]. Hotels are using it to greet customers upon entry [5]. Airports have adopted it for passport verification [6]. Even automobiles are now using it to enable drivers to access and start their vehicle [7].

The problem is FR algorithms can be vulnerable to spoofing attacks. Spoofing attacks are where an attacker presents a facsimile of an enrolled user, such as a picture, to gain unauthorized access. These attacks can be trivial to implement. Many people post photos of themselves on social media or professional websites, all the attacker needs to do is "replay" this likeness with the medium of their choice. This ease of attack combined with the wide-spread adoption of FR means there is clear value to developing a mass-implementable spoofing defense measure.

The National Institute of Standards and Technologies (NIST) helps clarify different types of attack presentations [8]. ISO 30107 shows that most attacks are generated from 2D imagery, cheap and easy to prepare. Examples include picture print-outs, display photos or videos, or creating a mask from paper or fabric. These attacks are potent for their production simplicity (cheap and only need a social-media photo). Attackers can easily fabricate variations of these instruments and expose detection methods vulnerabilities [8]. Relevant samples are shown in Figure 1 (see next page).

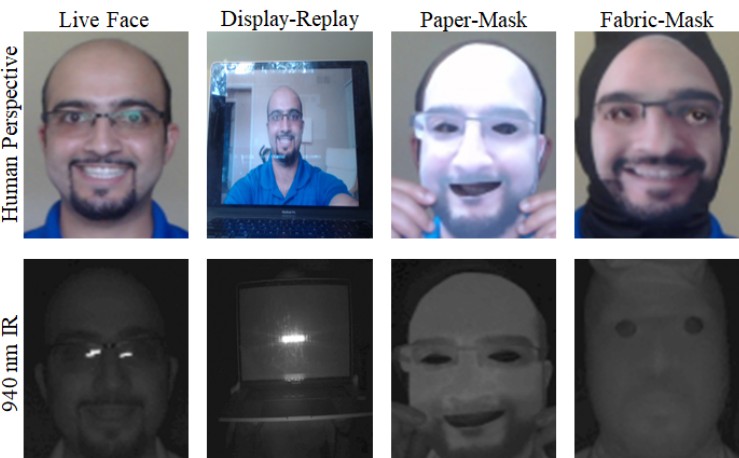

**Figure 1.** Juxtaposing the human and illuminated near-infrared perspectives. Spoofs have less texture variance and appear brighter to the infrared camera.

More complex attacks can be performed using 3D imagery. These include a latex or silica mask, printed-scaffolds and clay mannequins. While these are academically interesting, they are generally unrealistic. They are expensive to fabricate and require precise imaging, where most examples ironically involve the victim's participation in 3D scanning. For these reasons, both NIST and industry standards deem them out of scope [2,8].

This research proposes that near-infrared material spectroscopy can be used for robust facial presentation–attack–detection (PAD, also known as anti-spoofing). Historically, 3D technologies are typically best-in-class for PAD [9]. This can be completed using depth sensing [10] or using multi-frame deep learning networks [11]. The goal is not necessarily to supplant those methods, but address new goals of being monocular and single-frame. These are relevant due to many FR security systems being monocular, embedded systems.

Material spectroscopy is a process of identifying objects by shining a controlled light source and then characterizing the reflection [12]. This reflection is a function of the material's geometry and albedo; hence, an estimation of 3D structure can be made from textural features of the reflected light. Live faces are a combination of multiple surfaces with varying radii of curvature. A proof in the methodology shows how this must necessarily generate a distribution of varying frequencies (as a function of radius of curvature). Conversely, the spoofing attacks considered here are either flat that can be modeled as a simple convex surface. The methodology proof shows how this results in a simple low-frequency distribution. These distributions are further biased by the material, where spoofing materials are essentially uniform and very reflective (e.g., paper, fabric, and displays) and live faces are composed of skin, hair, eyes, etc., that vary in reflectivity.

The juxtaposition between the human perspective and illuminated near-infrared is shown in Figure 1. While the spoofs can look realistic to the human eye, they appear essentially washed out to the near-infrared perspective. This is because the spoof geometry results in low textural variance and the materials are extremely reflective. A mathematical surface-reflectance model is presented to show live faces should have predictably different reflectance distributions from their spoof counterparts. Fundamentally, it is the different in surface complexity and radii of curvature that drive reflectance variance.

One of the challenges of this research is the lack of near-infrared liveliness imagery. To address this, a dataset is collected using 30 diverse adults under varied pose, distance, and lighting. This process is repeated with four spoofing attacks inspired from NIST: paper-mask, spandex-mask, customized COVID mask, and display-replay. This results in a rigorous dataset that includes contrastive liveliness sets over several real-world noise factors, a fundamental contribution to the field. This yields approximately to 80,000 distinct frames, one of the paper's fundamental contributions.

The proposed material spectroscopy methodology is evaluated on the collected dataset. It is postulated that the spectroscopy process itself is the unique methodology, such that any reasonable texture algorithm should be able to discriminate liveliness with this imaging approach. This hypothesis is validated by benchmarking a panel of 13 deterministic [13–16] and deep learning [17–21] algorithms. Experimental results demonstrate excellent performance. The key contribution is the proposed-methodology enables a conventional MobileNetV3-based classifier [17] to outperform state-of-the-art networks selected from famous competitions [19–21]. This is not to discount the state-of-the-art (which are based off the smaller ResNet18 [22] encoder, but validate the methodology generates robust features. This reduces PAD to simply identifying a quality texture encoder.

These research contributions can be summarized as follows:

- Novel application material spectroscopy towards facial presentation–attack–detection;
- Novel near-infrared surface-reflectance mathematical model for facial-liveliness;
- Novel near-infrared data-collection of 30 diverse participants with NIST guided attacks that vary instrument, pose, distance, and lighting;
- Benchmarking the proposed methodology with 13 texture algorithms, enabling a conventional MobileNetV3 algorithm to achieve state-of-the-art performance.

The rest of the article is organized as follows. Section 2 presents related work associated with 2D inspired facsimiles. Section 3 describes the material spectroscopy theory with mathematical model of how live and spoof faces should reflect light. This model is the inspiration for using texture classification methods. Section 4 outlines the experiment design, including data-collection and algorithm implementation details. Section 5 presents the results, verifying the near-infrared reflectance can used for robust PAD. Section 6 discusses the work and outlines future research directions.

## 2. Related Works

Most facial-spoofing attacks are derived from 2D imagery due to complexity and cost [8]. These types of attacks, such as display a video or creating a mask from a paper-printout, can be performed with a trivial amount of production effort. The risks here are ever more present as people take more advantage of social media and share quality imagery online. This risk is highlighted by the presence of on-going facial presentation–attack–detection (PAD) challenges [23–25].

Ultimately, the goal is to identify methods that can deployed on mass. In principle, ultra-precise 3D sensing is the ultimate standard with PAD [26]. This research does not debate this, but rather notes the intended use-case (highly realistic 3D masks) is often unrealistic. In addition to being expensive (surveyed local costume companies charge between USD 5000 and USD 10,000), it requires a precise 3D model of the intended victim. While potentially feasible from 3D reconstruction methods, it is ultimately viewed as very unlikely. Once factoring in the cost and computational needs of such 3D sensing systems, it is more pragmatic to focus on mass-usability methods.

Hence, this related works places a strong emphasis on mitigating 2D-inspired spoofs. There is a clear need to develop methods that are mass-deployable. This survey identifies the relevant state-of-the-art, noting there is a literature gap with respect to monocular, single-frame algorithms. A discussion is presented on how the proposed material spectroscopy approach can address this.

### 2.1. Depth Sensing-Based Spoof Detection

Depth sensing is viewed as one of the fundamental technologies to recognize flat or simple 3D spoofs. Historically, depth is inferred by calculating the disparity between a stereo pair of cameras [27] (noting they can be mismatched [28]). These disparity maps, however, are often not precise enough for PAD. It is possible to improve accuracy using depth cameras [29,30], where a time-of-flight laser is used to achieve centimeter precision. With this said, a more popular trend is to use a structured-light emitter, projecting a known pattern, to precisely align the disparity map [31]. This light pattern is used to triangulate

points on the face and achieve millimeter depth precision. Many of the best works in the Face Anti-Spoofing Challenge competition utilized these depth methods [23].

Both of these technologies can be used to meet industry PAD requirements [26]. The problem at hand is accessibility. They are computationally heavy and can require extra perception hardware. This survey views these methods as robust, but not meeting the intended goals.

### 2.2. Motion-Based Spoof Detection

If depth technology is not available, a reasonable alternative is to pursue motion characteristics. Motion features describe the face by analyzing temporal differences or optical-flow. This work well for spoofs that have constrained facial structure, which intuitively should have fewer temporal components than real people.

Eye tracking is one of the simplest means to detect a still picture spoof. A simple blinking by tracking the landmarks around the eyes can be used to counter these attacks [32]. This is computationally cheap way to mitigate picture attacks without any impact to cost or user experience. That said, eye tracking should be always used in concert with more sophisticated methods, as it is trivial to defeat. A video replay inherently provides blinking, and an attacker can simply cut eye and mouth holes in a picture to make it a paper mask.

Heart-rate detection is another method that can be included in an ensemble anti-spoofing method. Blood flow across the face periodically pulses with the heartbeat [33]; each time the heart contracts, the veins across the face will similarly contract. This is detectable using frequency domain analysis to track the contractions [33]. It can be also detected by a shift in the green channel. For example, the hemoglobin blood absorbs green light, where a relaxed vessel full of blood inherently would reflect less green than a contracted one [34]. It is very difficult for even a highly realistic mask to produce a heartbeat, and a video will similarly pass these detection methods [35]. One could argue that an attacker needs simply to cut out a forehead patch of the mask and their own heart rate becomes present, but a more fundamental challenge is handling motion sensitivity. According to Li et al., the proposed method requires a 30-second time period to achieve a stable-signal [34]. While heart-rate verification can theoretically be accurate, it does not make sense in this use-case.

### 2.3. Texture and Color-Based Spoof Detection

Texture methods can theoretically detect facsimile artifacts. For example, printers can add distortion and quantize the colors of the intended face [36]. Likewise, both material and geometry can bias the way the spoof interacts with light; this can bias the distribution and, therefore, the perceived texture [36]. From a theory perspective, a well-scoped texture classifier should apply well to PAD.

One of the popular facial-texture descriptors are local-binary-patterns (LBP) [37]. The LBP describe the local relative rate of change, or gradient, for a given image patch. This is particularly useful to generally differentiate between faces; however, Chingovska et al. rather famously demonstrated LBP cannot differentiate liveliness on color imagery [13]. This is believed to be a shortcoming of using a passive RGB camera instead of an illuminated near-infrared camera.

Color distributions can also theoretically identify spoof production artifacts [16]. Wen et al. have demonstrated that ranked-channel histograms can in fact capture the color artifacts, but their results are only shown to work under static lighting. When reproducing their work, this research finds this approach to be too sensitive ambient lighting. This may also be another example of where using illuminated near-infrared can provide stability.

The facial spoofing challenges demonstrates that texture methods can be enhanced by the used of sophisticated deep learning networks [25]. There are some exceptional methods presented that introduce sophisticated feature modeling techniques. Examples include utilizing unique convolutional kernels to better improve texture features [19], projecting spoof cues using generative techniques [20], and even inferring depth maps from 2D images

using multi-task learning techniques [21]. These algorithms are promising and do very well on the competition datasets. The concern, however, is feature-translation. One general problem with presentation attacks is they are easy to vary. Should the algorithm over-fit to features relevant to that specific presentations in the training dataset, they will not generalize to minor variations introduced as the attacks evolve (an issue known with DeepFake detection [38]. It is acknowledged that one solution is to do cross-dataset evaluation for experimental verification. The goal of this paper is instead to propose a physics-informed approach, which should intuitively should generalize well while retaining the goals of robustness and efficiency.

### 2.4. Texture and Motion Fusion-Based Spoof Detection

Recent state-of-the-art methods emphasize combining texture and motion. In theory, both provide relevant features that are individually insufficient but their fusion can serve the goal. This can be performed very explicitly, such as take a sequence of texture maps. For instance, Shao et al. have demonstrated that by first taking the LBP of faces and then generating optical-flow maps [39]. This method is rather accurate on even 3D mask spoofs, but comes at significant computational complexity. Others have taken a slightly more elegant approach and use spatio-temporal networks, using temporal layers on a texture network to infer features in lieu of LBP pre-processing [40,41]. These methods are again accurate at significant computational complexity.

It is also an open question as to whether or not any temporal network is robust against introduced motions. Most evaluation datasets keep the spoof mask relatively still, such that motion generated by eye-blink or respiration is obvious. Introducing micro-motions by quickly shifting the mask around may prove challenging to differentiate. This attack vector is analyzed in this research.

### 2.5. Literature Opportunity: Near-Infrared Reflectance Patterns

This paper elects to build on the concept of 3D analysis through near-infrared spectroscopy. This research proposes that the differences in geometry (and albedo) are quantifiable in the reflectance patterns. The concept is using illumination to infer shape has been used biomedical applications [42]. The key contribution here is historical methods assume multiple perspectives to determine orientation [42,43]. Furthermore, near-infrared is considered ideal for identification due to its general robustness towards ambient-light [44–46]. This proposal presents a mathematical model that proves a single-frame is sufficient for facial-PAD, as faces behave similarly to Lambertian-surfaces [47].

It is important to acknowledge there are preliminary works that use reflectance intensity for liveliness. Zhang et al. have verified that skin does in fact have a notably lower albedo than paper at 850 nm (another NIR wavelength), noting the reflectance is also a function of distance [48]. Their research, however, does not propose a method to accurately estimate the distance nor control for environmental factors. Chan et al. have introduced that the concept of a flash to compensate out the ambient [49]. This is an improvement, but it does not address distance. Neither research sufficiently analyzed impact of pose nor other types of spoofs.

Lastly, Agarwal et al. demonstrate a combination of near-infrared and thermal-infrared can yield excellent results [50]. This methodology is deemed best in class, as it is intuitive that is exceptionally difficult to fake thermal radiation patterns. That is to say even if the spoof can conduct heat from the face (or be heated to skin temperatures a priori), it is highly difficult to mimic the blood vessel radiation patterns. Their work is highly relevant, but requires the additional thermal camera for robustness. This paper extends those findings by only requiring the near-infrared camera.

Hence, this research presents a novel means of efficient facial presentation–attack–detection using reflected NIR texture. It addresses noise factors not considered in prior works. In addition, it introduces additional attack vectors in the form of video-replay and fabric masks and investigates effectiveness of the proposed method for these new attacks.

### 3. Near-Infrared Reflectance Methodology

This paper proposes a material spectroscopy approach towards facial presentation–attack–detection (PAD). The hypothesis is as follows: facial geometry can be characterized by shining a near-infrared light and measuring the reflectance patterns. This is modeled mathematically, where surface-reflectance equations are used to demonstrate there are necessarily differences between live and spoof faces. It is then proposed that texture methods can optimally do the PAD classification. Note that the novelty here is the spectroscopy methodology—not a specific algorithm. This process of shining a near-infrared light and then characterizing liveliness using a texture classifier is how robust PAD is achieved in a monocular, single-frame fashion.

The expected reflectance distributions for live and spoof faces are illustrated in Figure 2. Faces reflect the near-infrared illumination as a function of their geometry and albedo (reflectivity). Live faces have convex and concave components, which results in higher frequencies and varied reflectance distributions. Simple spoofs, conversely, are generally a single convex surface and often result in a simple, low-frequency distribution. This difference in frequency content will be used to characterize geometric complexity.

Live-Face: Convex and Concave Reflections

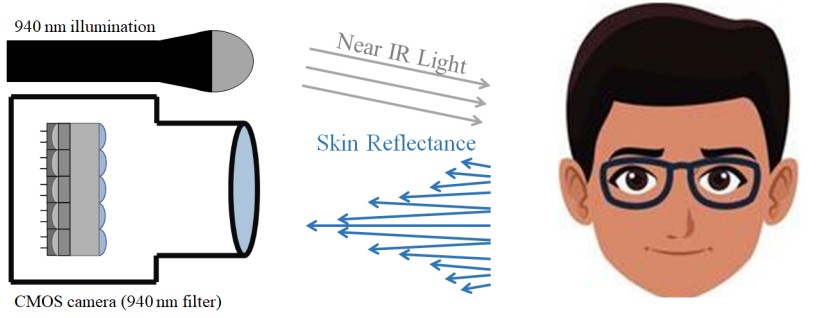

Simple-Spoof: Primary Convex Reflection

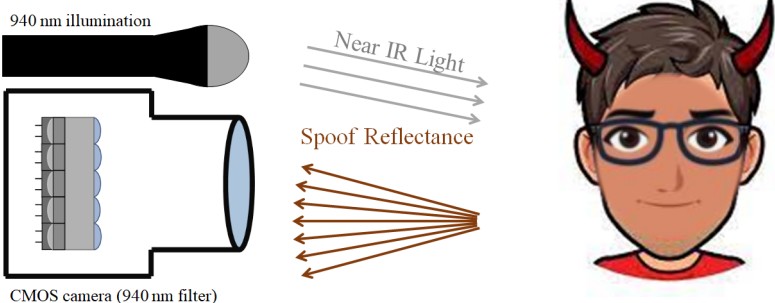

**Figure 2.** Using material spectroscopy principles for liveliness. Live people have complex geometry with highly variant reflectance patterns. Simple spoofs conversely have relatively uniform reflectance patterns. These patterns can be classified using texture methods.

Material albedo can also play a key role in classifying liveliness. Spoofs are designed to look realistic to the human eye, but the concept of "color" may not necessarily translate to other spectra. Recall how the spoofs look notably more reflective in Figure 1 are more reflective to near-infrared than human skin [48]. This is because the common spoofs materials (e.g., paper, fabric, latex, and glass) are more reflective to near-infrared than human skin [48,51–53]. Note that this research recommends 940 nm infrared because it has minimal contribution from solar radiance [54] or household LED light sources [55]. This helps isolate the reflectance signal.

### 3.1. Facial Reflectance Modeling

The facial reflectance can be modeled by a combination of convex and concave Lambertian surfaces. A simple spoof, such as paper printout, is largely flat with slight convexity when bent to the face. This can be modeled as a simple convex sphere. A live face has convex and concave portions. For example, the nose is convex and the eye sockets are concave. This can be modeled as a primary sphere that has secondary convex and concave spheres. When illuminated, these secondary surfaces bias the reflectance distribution, adding scattering effects. These effects are proven to necessarily introduce additional frequency terms using surface reflectance physics. This frequency content is can be detected using texture methods to determine liveliness.

A few assumptions are made for simplicity. First, this proof is completed assuming only two-dimensions. Because the models are based off spheres, there is a uniform radius across all points. This means the rate of change in reflectance with respect to the horizontal axis, $\theta$, is identical to the rate of change with respect to the depth axis, $\rho$. Demonstrating the surface impacts on circles should intuitively hold true for spheres. Furthermore, not all the components on the face are Lambertian. For example, hair and spoof materials (e.g., fabric) do not perfectly reflect light. This simplification is justified in the experimental results, which demonstrates faces sufficiently generate the expected reflectance distribution.

Three Lambertian surface-models are visualized in Figure 3: simple-convex (A), secondary-convex (B), and secondary-concave (C). Note that these models are only relevant for the surface angle range corresponding with positive y values; that is to say, $\theta$ is bound between the intercept angles $\theta_s$ and $\pi - \theta_s$. Furthermore, the models are valid only for the indicated type of surface. Model A is always valid, and Models B and C are only valid for angles bound within the secondary-surface (elsewhere defaulting back to Model A).

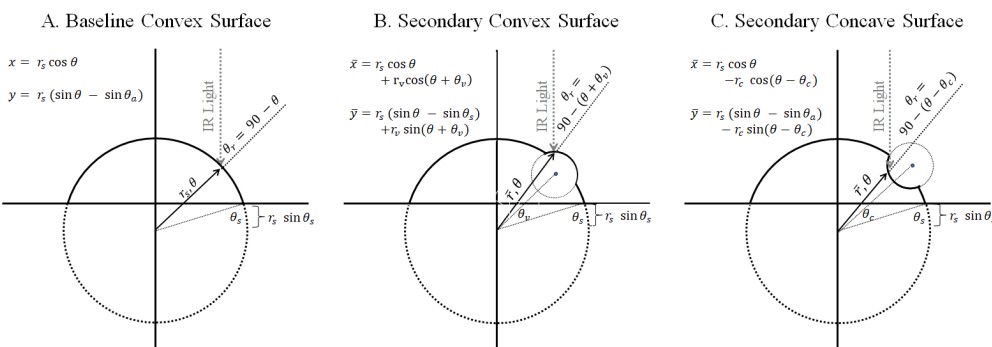

**Figure 3.** Facial surface reflectance models. Model **A** is the baseline convex surface. Model **B** introduces a secondary convex surface, increasing reflectivity and phase. Model **C** introduces a secondary concave surface, decreasing reflectivity and phase.

Lambertian surfaces reflect diffuse light as a function of the source intensity, angle from the surface normal and albedo [56]. The surface reflection for an active illumination source is calculated in Equation (1). $SR$ is the surface reflection, $\frac{\rho_d}{\phi}$ is the albedo (reflectivity coefficient), $I$ is the infrared intensity, and $\theta_r$ is the reflectance angle between incident ray and surface normal. Note that for now the facial albedo is assumed to be a constant dependent associated with the primary material, such as skin, paper, glass, etc. This introduction of sub-components will be accounted for later.

$$SR = \frac{\rho_d}{\pi} I_r \times \cos(\theta_r). \tag{1}$$

The surface geometry directly impacts the infrared intensity. By definition, light waves decrease quadratically with the source-distance. This is generalized in Equation (2) by

assuming the $I_0$ is the light intensity, and the source-distance is difference between from the further part of the face, $d_{face}$, and surface height $y(\theta)$.

$$I_r = \frac{1}{4\pi} I_0 \times \frac{1}{(d_{face} - y(\theta))^2}. \tag{2}$$

Calculating the surface height and normal vector can also be modeled as a function of surface angle and radius of curvature. The surface angle $\theta$ is the angle with respect to the $x$ axis. The surface radius of curvature $r_s$ is radius of the tangential circle. Note that the circle focal point may be offset on the $y$ axis. A perfectly circle surface by definition would have focal point at the origin. However, flatter surfaces would have a longer radius of curvature and therefore be an offset on the $y$ axis.

Assume that the surface angle which intersects the $x$ axis can be defined as $\theta_s$. This means that the tangent circle focal point is then offset by $r_s \sin \theta_s$. Recall the three Lambertian-surface models from Figure 3. Model A is simplest, where the height is the $x$ axis projection and the surface-normal is the surface angle. Introducing secondary surfaces, however, now it requires projecting a translation as a function of to the secondary surface radius and angle to the secondary-arc focal point. Model B presents a constructive interaction where the secondary convex arc has focal point at $r_v, \theta_v$. The surface height is increased by the projected difference between the original arc and sub-arc focal point, and the surface-normal is increased by $\theta_v$ in phase. Model C makes a similar assumption that the concave-arc focal point is the location $r_c, \theta_c$. This time the translation is destructive; the secondary-surface height is decreased by the projected difference between the original arc and sub-arc focal point, and surface-normal is decreased in phase by $\theta_c$.

$$d(\theta) = \begin{cases} d_{face} - r_s \cdot (\sin(\theta) - \sin(\theta_s)), & \text{Model A} \\[2ex] \begin{aligned} & d_{face} - r_s \cdot (\sin(\theta) - \sin(\theta_s)) \\ & \quad - |r_s - r_v| \cdot \sin(\theta + \theta_v), \end{aligned} & \text{Model B} \\[2ex] \begin{aligned} & d_{face} - r_s \cdot (\sin(\theta) - \sin(\theta_s)) \\ & \quad + |r_s - r_c| \cdot \sin(\theta - \theta_c), \end{aligned} & \text{Model C} \end{cases}. \tag{3}$$

Surface reflectance is a function of distance as applied in Equation (3). More specifically, this distance is a function the furthest face distance $d_{face}$, surface radius $r_s$, surface angle $\theta_s$, and surface bounding angle $\theta_s$ (noting secondary surfaces also have radii $r_v, r_c$ with offset angles $\theta_v, \theta_c$). Intuitively, one can see the radius term, $r_s$, becomes dominant for flatter surfaces. This will inherently reduce the valid surface angles to be closer to the $y$-axis (i.e., $\frac{\pi}{2}$), fundamentally acting as a low-pass filter. This is particularly relevant for simple spoofs, such as paper mask and display-replay, that are generally flat. That is to say the simple spoofs should only have low frequency content.

$$I(\theta) = SR(\theta) + I_{amb}. \tag{4}$$

For completeness, the surface reflectance also needs to factor the ambient light. This is simply a summation of the infrared surface reflectance and light present at the surface $I_{amb}$, as applied in Equation (4). Ambient here is simplified to be diffuse (i.e., it is approximately uniform across the image).

These equations can be combined for an explicit total reflectance equation. Adding secondary surfaces necessarily generates new frequency content in the reflectance profile as applied in Equation (5) (see next-page). This, in theory, should generate more variance in the distribution, a behavior which the texture classifier should easily identify. Note here the albedo, source intensity and ambient are constants and should not impact the light variance. Furthermore, the illumination source angle, $\theta_I$, is visualized as 90 degrees in the

surface models figure but only acts as a phase shift. Position should theoretically matter less than the geometry of the object.

$$I(\theta) = \frac{\rho_d}{4\pi^2} I_0 \times
\begin{cases}
\frac{1}{(d_{face}-r_s(\sin(\theta)-\sin(\theta_s)))^2} & \\
\quad \times \cos\left(\theta_I - \theta\right) + I_{amb}, & \text{Model A} \\
& \\
\frac{1}{(d_{face}-r_s(\sin(\theta)-\sin(\theta_s))-|r_s-r_v|\sin(\theta+\theta_v)))^2} & \\
\quad \times \cos\left(\theta_I - (\theta+\theta_v)\right) + I_{amb}, & \text{Model B} \\
& \\
\frac{1}{(d_{face}-r_s(\sin(\theta)-\sin(\theta_s))+|r_s-r_c|\sin(\theta-\theta_c)))^2} & \\
\quad \times \cos\left(\theta_I - (\theta-\theta_c)\right) + I_{amb}, & \text{Model C}
\end{cases}. \tag{5}$$

To verify the geometry is more important than position or light intensity, the surface reflectance derivative can be taken. Complex surfaces have significantly more sinusoidal terms in the derivative (not shown for space reasons but is intuitively obvious). This necessarily means there is a more variance across the surface reflectance distribution, associated with the key facial structures.

### 3.2. Liveliness Hypothesis

Putting these equations together yield a few interesting findings. First and foremost, the object flatness acts as a low-pass filter. Recall how the surface angle is bound by the intersection angle $\theta_s$; flatter objects have a longer radius of curvature and, therefore, a reduced range of surface angles. This is particularly relevant for simple spoofs, which can be largely flat by design (e.g., paper mask or display-replay). Conversely, live faces are sufficiently curved to introduce a broader range of frequencies. This phenomenon is enhanced with complex geometries. Secondary convex and concave surfaces generate more variance in reflectance profile, with new inflection points caused by changing concavity (e.g., nose transitioning to eyes). Even fabric masks fail to fully capture these secondary surfaces, behaving more like a single convex surface. This behavior is indicated by Equation (5) (and implied by the derivative, not shown), visualizing the different profiles.

### 3.3. Classification Methodology

Texture classifiers should be optimally situated to discriminate these reflectance profiles. In this regard, the methodology is not about a particular algorithm, but rather the material spectroscopy process. This claim is verified in the experiment by benchmarking 10 relevant texture classifiers on the evaluation dataset. The goal is to demonstrate the spectroscopy approach generates very robust features, where the application developer can optimize performance and run-time.

## 4. Presentation–Attack–Detection Experiment

A set of experiments are designed to evaluate effectiveness and robustness of the proposed material-spectroscopy-based facial presentation–attack–detection (PAD) approach. Live actors and their corresponding spoofing attacks (paper mask, spandex mask, customized COVID mask and display-replay) are evaluated under various lighting and positions. These attacks are selected based off NIST ISO 30107 Levels A and B, facsimiles generated from photos [8]. It is proposed that near-infrared illumination introduces differences in the facial reflectance due to surface geometry and albedo, which can be robustly classified using texture methods. This is verified by benchmarking a panel of 10 relevant deterministic and deep-learning algorithms. Note that the evaluation requires collecting a novel dataset; current open-source datasets do not contain sufficient illuminated near-infrared imagery. Furthermore, this is the first dataset to introduce fabric and customized COVID masks.

### 4.1. Dataset Collection

Dataset liveliness presentations are given in Table 1. This collection is a robust contribution both due to its spectral content and strong variance in attributes. In total, 30 diverse adults are imaged, representing gender (20 males and 10 females), 6 ethnicity groups and ages ranging between 20 and 65. The participants are imaged with live and spoof presentations under the varied perspectives. Imaging is performed with a 5 mega-pixel FLIR Blackfly monochrome camera [57] employing a 940 nm filter with matching illumination.

**Table 1.** Experiment liveliness presentation matrix. All participants are imaged under every matrix combination with the exception of sun-diffuse (all spoof with some live, indicated by *).

| Presentation | Ambient | Distance (Meters) | Yaw (deg) | Pitch (deg) |
|---|---|---|---|---|
| Live (30) | Dark, Lights, Sun * | [0.5, 1.5] | [−45, 45] | [−15, 15] |
| display-replay (30) | Dark, Lights, Sun * | [0.5, 1.5] | [−45, 45] | [−15, 15] |
| Paper Mask (30) | Dark, Lights, Sun * | [0.5, 1.5] | [−45, 45] | [−15, 15] |
| Spandex Mask (30) | Dark, Lights, Sun * | [0.5, 1.5] | [−45, 45] | [−15, 15] |
| Face-Print Covid Mask (30) | Dark, Lights, Sun * | [0.5, 1.5] | [−45, 45] | [−15, 15] |

The base condition is the participant themselves sitting in the camera lab, looking at the camera and performing several head nodding motions. This is approximately distance ranging from 0.5 to 1.5 m, yaw ranging from −45 to 45 degrees and pitch ranging from −15 to 15 degrees. This process is performed in laboratory dark conditions (940 nm illumination only), laboratory light conditions (940 nm illumination with all lab lights turned on), and diffuse outdoor sun conditions (940 nm illumination with sunlight diffused by glass). This process is then repeated for all spoofing attacks: display-replay, paper mask, spandex mask and customized COVID mask. All images are pre-processed with the Retina Face [58]. Face crops are re-scaled to either 225 × 225 for deterministic algorithms or appropriate deep learning network input size.

In total, there are approximately 80,000 unique frames for evaluation (62,000 laboratory and 18,000 exterior). Note that the sun-diffuse condition is a retro-active add-on; all spoofing attacks are presented, but only a subset of the original participants is still available for imaging. Furthermore, the exterior procedure is optimized to capture the attributes with fewer frames. To avoid data bias, the evaluation is first conducted using only laboratory data and then using the full dataset.

Sample dataset face-crops are shown in Figure 4 (see next-page). This visualization shows relevant samples imaged using illuminated near-infrared juxtaposed with their human perspective. While these spoofs look generally realistic to the human eye (which is verified using FaceNet [1] embeddings comparison, not shown for space), there are obvious differences in the near-infrared perspectives. Both in a lab and outdoors, the spoofs have reduced textural variance due to the simplified geometry. These texture effects are enhanced by the material albedo, where the paper and fabric are notably higher than skin (causing these spoofs to generally appear lighter). Furthermore, the display-replay attacks do not even show up to the near-infrared camera (they do not emit in IR [55]). This is viewed as a passive form of anti-spoofing. Note that all evaluation (training and testing) is performed using only the illuminated near-infrared imagery. The human-perspective is shown only to visualize how the spoofs look sufficiently realistic.

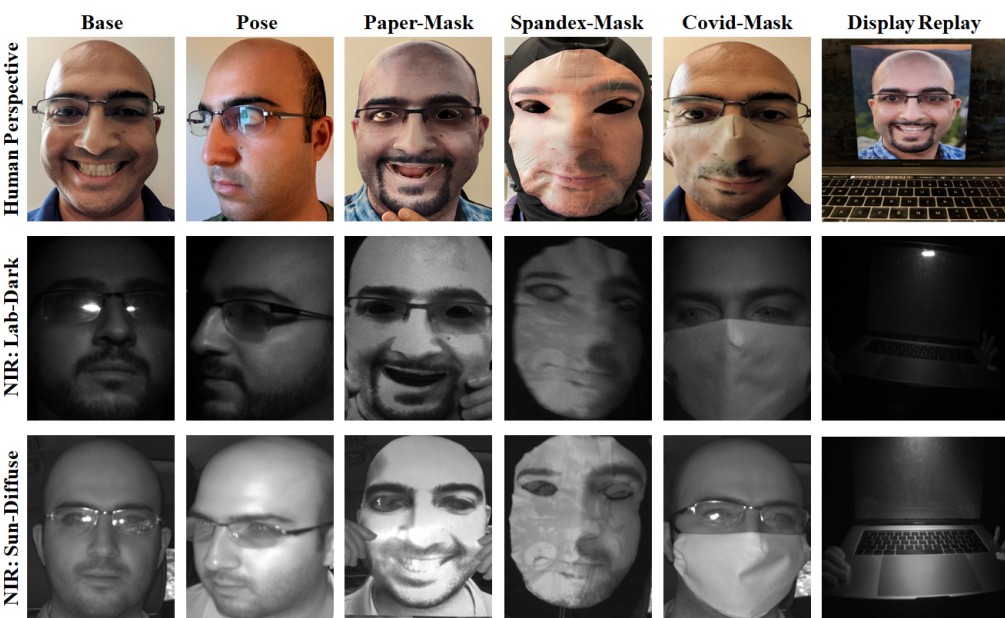

**Figure 4.** Data collection visualization. This matrix juxtaposes the different types of perspectives captured, noise and spoof attacks (noting that all spoofs also are present with varied pose and distance). This perspective helps visualize the benefits of using illuminated near-infrared versus RGB. Spoofs can be made to easily match the human-perspective, but look like a low-pass filter is applied under the illuminated near-infrared.

### 4.2. Texture Classifiers for Benchmarking

To evaluate effectiveness of the proposed infrared reflectance pattern-based PAD, a panel of deterministic [13–16] and deep learning [17–21]. These are selected based upon historical relevance in image classification. A brief overview of the texture classifiers is provided in the following.

#### 4.2.1. Deterministic: Local Binary Patterns

The first method is the local binary pattern (LBP) [59]. LBP are texture descriptors computed by a binary kernel. They are popular in facial identification, and are demonstrated for liveliness by taking a multi-block approach [13]. For PAD, the face-crop is first described with the LBP feature using the science kit image toolbox [60] and analyzed as a histogram feature (128 bins) using the science kit learn toolbox [61].

#### 4.2.2. Deterministic: Discrete Cosine Transform

The second method is the discrete cosine transform (DCT) [62]. DCT is frequency analysis tool similar to the fast-Fourier-transform, but generally viewed as more explainable. It has historical uses in face recognition [14] and can theoretically discriminate the liveliness classes by reflectance frequency content. For PAD, the face-crop is first described by the DCT algorithm using the science python toolbox [63], where the frequency terms are extracted via zig-zag pattern for classification. principal component analysis (PCA) is used to select 128 features.

#### 4.2.3. Deterministic: Ranked Channel Histograms

The third method is the ranked channel histograms (RCH) [64]. Others have shown that the distribution of color channels can be used for liveliness, where they used principal component analysis to identify the key components from each channel's histogram. This approach can similarly be used to describe the variance of infrared light reflection, and, therefore, correlate with the texture. For PAD, the face-crop is described by its histogram

using the science kit learn toolbox [61] and then the 25 primary bins as determined by PCA are used for classification.

### 4.2.4. Deterministic: Random Fourier Series

The fourth method is the random Fourier series (RFS), also known as random kitchen sinks [15]. This feature is theoretically similar to DCT; however, instead of explicitly calculating relevant frequency features, they are generated as a sum of randomized series. Prior applications have shown promise using random features, though there is risk a poor set of frequency series are selected. For PAD, the face-crop is first described by the RFS algorithm using the science kit image toolbox [60], where the output is then unraveled into an array for classification. Principal component analysis is used to select 128 features.

### 4.2.5. Deep Learning: Baseline Image Classification Networks

The fifth method introduces deep learning (DL) starts with baseline image classification networks. This method uses two popular image classification encoders and transfer learns them for PAD. The face-crop is described by a convolutional neural network (CNN) feature encoder then classified using a 128-neuron feature fully connected layer and 2-neuron classification layer. Feature space is further evaluated by employing the MobileNetV3 encoder [17] for an efficient network and the InceptionV3 encoder [18] for a robust network. Both encoders are pre-trained on ImageNet [65] and fine-tuned on the training dataset for evaluation. Training and evaluation is conducted using the PyTorch Lightning toolbox [66].

### 4.2.6. Deep Learning: Central Difference Network

The sixth method employs the central difference convolutional network (CDCN, simplified to central difference net) [19]. This is a state-of-the-art network that employs an attention module (i.e., the central difference convolution) to better capture texture features and won the CVPR 2020 facial anti-spoofing challenge [19]. The face-crop is described by a modified ResNet18 [22] that encodes spatial-attention features at multiple levels, and uses them for a classification binary-map. ResNet18 encoders is pre-trained on ImageNet [65] and fine-tuned on the training dataset for evaluation. Training and evaluation is conducted using the PyTorch Lightning toolbox [66].

### 4.2.7. Deep Learning: Spoof Cues Network

The seventh method employs the learning generalized spoof cues network (LGSCN, shortened to spoof cues net) [20]. This an interesting generative network that projects the expected spoof cues and compares them with the input for liveliness classification. It is actually designed for DeepFake detection, achieving first place on the FaceForensic [67] benchmark in 2021 [20], but intuitively can be applied for PAD. The face-crop is described by the ResNet18 encoder [22] and then projected using generative layers into a spoof cue. This cue is compared against the input using an auxiliary classified to determine liveliness. Training and evaluation is conducted using the PyTorch Lightning toolbox [66].

### 4.2.8. Deep-Learning: Dual-Branch Depth Network

The eighth method employs the dual-branch meta-learning network (DBMLN, simplified to dual-branch depth net) [21]. This is a multi-task learning network that utilizes liveliness embedding and depth-estimation tasks to intuitively incorporate 3D features into the embeddings. The result is excellent cross-dataset evaluation performance as validated by TIFS [21]. The face-crop is described by the ResNet18 encoder [22] and a liveliness score is combined from both embedding and depth tasks. Because this evaluation dataset does not contain depth maps, a pre-trained model is used directly from the Github (utilizing competition datasets) then fine-tuned on the training dataset. Training and evaluation is conducted using the PyTorch Lightning toolbox [66].

### 4.3. Texture Method Evaluation

To highlight the utility of infrared reflectance pattern-based PAD, a series of texture features are evaluated for performance evaluation. The deterministic features are generated using texture descriptors and then classified using a random forest classifier [68] via the science kit learn toolbox [61]. The deep-learning features are classified using a fully connected layer. All methods are evaluated using binary classification, a live face or a spoofed face. Given some of the deterministic features do over-fit with the COVID mask spoof (which is hybrid of both classes), an ensemble is also evaluated for the 3D classes (deterministic features only). This ensemble is a majority vote for three binary classifiers trained to each spoofing attack.

Note that these methods are generally known for other texture classification applications. The novelty here is the application of illuminated near-infrared imaging for PAD via material-spectroscopy. Recall from the related works prior texture methods employ sophisticated ensembles [25]. These algorithms are simple to compute.

All selected texture classifiers are trained using stratified cross-validation. Test values are the average of 10 randomized training and testing evaluations (80:20 participant ratio to maintain independence). Classification evaluation metrics are nominal presentation classification error rate (NPCER), attack-presentation classification error rate (APCER) and the average classification error rate (ACER, average of NPCER and APCER). NPCER is synonymous with the false-rejection-rate of live people; APCER is synonymous with the false-acceptance rate of spoofing attacks.

### 4.4. Research Limitations

Facial spoofing is inherently an evolving field, where each countermeasure will eventually be exposed by attackers. The attack presentation methods selected are designed to represent the most common attacks with relevant noises, but are inherently not inclusive. A general pragmatic approach to algorithm development would be to verify requirements on this evaluation dataset, then make updates based upon penetration testing.

## 5. Experimental Results

The proposed spectroscopy methodology is evaluated by imaging 30 people and their corresponding spoofs under near-infrared illumination. Based upon the surface reflectance modeling, it is hypothesized texture classifiers can robustly characterize facial liveliness based upon geometric and albedo differences. This hypothesis is evaluated here by benchmarking a panel of texture classifiers on the novel spectroscopy dataset, first using laboratory data then including exterior data (see Section 4 for details). The evaluation results presented next validate the hypothesis, demonstrating excellent facial presentation–attack–detection (PAD) accuracy.

Laboratory experimental results (dark and lights on) are given in Table 2 (see next-page). The first result column is the overall accuracy on the entire dataset. The second result column is the nominal presentation classification error rate (NPCER, essentially false-rejection rate of live people). The remaining columns shows the attack-presentation classification error rate (APCER, essentially false-acceptance-rate of spoofs). Note that the display-replay attacks are all represented as not-detected (ND). This is because near-infrared cameras do not see the RGB face emitted from the display.

This demonstrates that even deterministic texture descriptors can differentiate the liveliness materials. The local binary pattern algorithm (LBP, indicated by the †) is the optimal solution for its strong standalone accuracy with minimal class bias. The discrete cosine transform and ranked channel histograms similarly show good general robustness; however, it appears the COVID mask class causes some bias towards false-rejections. This bias can be addressed through the use of the 3D ensemble. Note how not only the average precision improves, but the error rates are better distributed.

**Table 2.** Facial presentation–attack–detection experiment results: lab conditions. † indicates optimal deterministic algorithm. ‡ indicates optimal deep learning algorithm.

| Algorithm | ACER | NPCER | APCER: | | | |
| --- | --- | --- | --- | --- | --- | --- |
| | | | Paper | Spandex | COVID | Display |
| Det: LBP † | 3.6% | 4.9% | 6.4% | 0.2% | 0.0% | ND |
| Det: DCT | 9.8% | 17.6% | 1.0% | 5.2% | 0.0% | ND |
| Det: RCH | 14.9% | 23.3% | 11.5% | 4.6% | 0.0% | ND |
| Det: RFS | 50.1% | 99.9% | 0.4% | 0.1% | 0.1% | ND |
| Det 3D-Ensemble: LBP | 1.7% | 0.9% | 7.7% | 0.0% | 0.0% | ND |
| Det 3D-Ensemble: DCT | 3.8% | 3.8% | 2.2% | 8.4% | 0.5% | ND |
| Det 3D-Ensemble: RCH | 6.7% | 8.1% | 10.2% | 0.4% | 5.5% | ND |
| Det 3D-Ensemble: RFS | 24.8% | 31.0% | 17.0% | 17.0% | 22.0% | ND |
| DL Base: MobileNetV3 ‡ | 0.2% | 0.2% | 0.3% | 0.0% | 0.1% | ND |
| DL Base: InceptionNetV3 | 0.3% | 0.4% | 0.5% | 0.0% | 0.0% | ND |
| SOTA: Central-Difference Net | 0.5% | 0.4% | 1.6% | 0.0% | 0.0% | ND |
| SOTA: Spoof-Cues Net | 1.0% | 0.5% | 2.9% | 0.1% | 1.4% | ND |
| SOTA: Dual-Branch Depth Net | 1.3% | 0.6% | 3.2% | 0.1% | 2.2% | ND |

The one poor performer is the random Fourier series. It is essentially an automatic-rejection tool as a single classifier, and still inconsistent in the 3D ensemble. It is postulated that the frequencies selected are outside of the realistic values for surface-angle $\theta_s$. This means that even when using principal component analysis, the random features are potentially for liveliness.

The best in class performance is achieved using deep learning. Surprisingly, the best performer is actually the base network with MobiletNetV3 encoder [17] (indicated by the ‡). While counter-intuitive, this is hypothesized for a couple of reasons. First, the state-of-the-art networks introduce sophisticated feature-generation methods, tailored to the competition datasets. In this scenario, the material spectroscopy process inherently well separates the liveliness classes. Hence, this imaging approach simplifies the problem to picking the optimal encoder. Note that the competition networks are also built off the smaller ResNet18 encoder [22], which is efficient, but not particularly potent on ImageNet. This enables the base networks (both MobileNetV3 [17] and InceptionV3 [18]) to better discern the attacks by simply having more robust encoders.

Delving further into the networks, a few hypotheses can be made. The texture features here are relatively low-dimensional; this is both presented in the mathematical model and observable in the dataset sample images. This suggests that the MobileNetV3 encoder [17] is sufficient due to its simple parameter space, whereas the more powerful InceptionV3 encoder's high dimensional features [18] are not applicable. This explains why the results are essentially tied with the best network (noting there could also be slight over-fitting to high dimensional noise). Furthermore, this aligns well with why the central-difference net [19] is the best of the state-of-the-art performers. While it also is built off ResNet18 [22], the convolutional kernel is designed to improve low-dimensional texture features. This aligns well with the mathematical model and is roughly in line with the base networks.

In addition to encoder limitations, there are some potential biases introduced in the other state-of-the-art networks. For example, the dual-branch depth network [21] results are generally good but surprisingly the worst of the deep learning. This can be easily rationalized as a consequence of not having depth-maps for training in this dataset. While a pre-trained version is transfer-learned from the competition datasets, this likely is not enough to retain the depth information when fine-tuning the embedding task.

Additionally, the spoof cues network [20] is designed to project a spoof perspective and compare it against the original image. Intuitively, the distance from the projection should be greater when there is a live image presented versus spoof. In practice, there is a potential methodology limitation. Because it is always projecting spoofs, it may incorrectly learn generative features when the input is a spoof (e.g., not learning what constitutes a

live face). An interesting adaption would be to project both live and spoof perspectives in a multi-task-learning design, then do a contrastive difference with the input.

The spectroscopy approach is also robust to exterior sunlight as given in Table 3. Most algorithms perform within statistical noise of the laboratory conditions (with the exception of the random Fourier series). In some cases, the results actually improve. This is believed to be a result of the diffuse sun-load adding constructively with the illuminator. Recall from the proof the ambient light is essentially a zero-frequency term in Equation (5). This helps with general image quality, as the additional light improves the overall exposure, without impacting the texture features.

**Table 3.** Facial presentation–attack–detection experiment results: lab and exterior sun conditions. † indicates optimal deterministic algorithm. ‡ indicates optimal deep learning algorithm.

| Algorithm | ACER | NPCER | APCER: | | | |
| | | | Paper | Spandex | COVID | Display |
|---|---|---|---|---|---|---|
| Det: LBP † | 2.6% | 4.0% | 3.5% | 0.1% | 0.1% | ND |
| Det: DCT | 7.9% | 13.5% | 1.2% | 5.6% | 0.0% | ND |
| Det: RCH | 9.4% | 15.9% | 8.3% | 0.0% | 0.1% | ND |
| Det: RFS | 49.7% | 7.5% | 91.4% | 96.8% | 87.6% | ND |
| Det 3D-Ensemble: LBP | 1.6% | 2.4% | 1.9% | 0.2% | 0.1% | ND |
| Det 3D-Ensemble: DCT | 4.7% | 2.9% | 5.5% | 13.3% | 0.4% | ND |
| Det 3D-Ensemble: RCH | 6.7% | 10.6 % | 8.0% | 0.4% | 0.3% | ND |
| Det 3D-Ensemble: RFS | 50.0% | 99.9% | 0.1% | 0.2% | 0.1% | ND |
| DL Base: MobileNetV3 ‡ | 0.8% | 1.3% | 0.3% | 0.0% | 0.2% | ND |
| DL Base: InceptionNetV3 | 0.3% | 0.4% | 0.5% | 0.0% | 0.0% | ND |
| SOTA: Central-Difference Net | 1.1% | 1.8% | 1.3% | 0.0% | 0.0% | ND |
| SOTA: Spoof-Cues Net | 1.5% | 1.6% | 2.2% | 0.1% | 1.3% | ND |
| SOTA: Dual-Branch Depth Net | 1.9% | 2.0% | 2.8% | 0.0% | 1.9% | ND |

The outdoor environment, however, does add additional noise. This is visualized in the form of degraded gamma and pepper noise in Figure 4. The introduction of low-frequency noise likely interferes with the deep-learning networks to be discern the liveliness texture features, which causes a general drop in performance except for the base InceptionV3 network [18]. Here, the benefits of the sophisticated encoder becomes present, as it is definitively the most accurate network. Still, the base MobileNetV3 network [17] is indicated as the most optimal for its better combination of similar accuracy with notably better efficiency. Perhaps more interesting is the ranked channel histograms remain stable across lighting conditions. This particular descriptor is hypothesized to be the most light sensitive, but the relative distribution remains sufficient for liveliness.

These results suggest either the LBP or MobileNetV3 algorithm should be deployed, depending upon the inference platform. If it is a CPU-based platform, the LBP algorithm is sufficiently robust and can be completed in real time. On the other hand, if a deep learning accelerator is available, the MobileNetV3 algorithm is a better choice. It is extremely robust and can be run with negligible latency to the user. Note that while the 3D ensemble can address bias from the COVID mask classes (which are essentially a hybrid of live and spoof), it generally is not worth the extra computational cost.

## 6. Discussion

This paper demonstrates that near-infrared reflectance patterns can enable robust monocular facial presentation–attack–detection (PAD). PAD is a key part to securing face recognition, mitigating spoofing attacks without falsely rejecting authorized users. Common spoofing attacks include printed pictures, displaying photos or videos, and fabric masks. State-of-the-art methods historically use 3D features to detect these attacks; however, those methods require either extra depth sensors or complicated multi-frame algorithms.

This research builds off the theory that 3D-geometry features are useful and proposes a novel way to utilize them in a monocular, single-frame fashion.

The contribution is the proposed material spectroscopy mathematical model and demonstration of state-of-the-art performance. Live and spoof faces have predictably different reflectance patterns when illuminated with near-infrared light. This hypothesis is modeled mathematically, where surface reflectance is modeled as a function of geometry and albedo. This reflectance behavior implies the frequency composition is sufficient to discern the facial surface, and, therefore, discern liveliness. This hypothesis is then validated by benchmarking a panel of 13 texture classifiers on the prepared dataset. This dataset is extremely diverse, introducing multiple attack vectors over varied light and position for a total of 80,000 unique frames.

As anticipated, most of the algorithms classify liveliness extremely well. Both deterministic and deep learning algorithms achieve quality results. The key finding is this material spectroscopy process enables a traditional MobileNetV3 network [17] to achieve 0.8% average classification error rate—outperforming the algorithms selected for their competition performance. This is not to say that the traditional network is superior (as it has a larger encoder than the competition networks), but validate the methodology generates robust features. This simplifies algorithm design while achieving robust, efficient PAD.

Future works aim to expand the dataset noise factors and verify features translate across datasets. This collection is designed to verify the proof, meaning the ambient light is structured to be diffuse. Expanding the environment noise factors to include point-sources and shadows would represent challenging real-world scenarios. Additionally, a fundamental claim is this imaging approach inherently generates reliable liveliness features. Cross-dataset evaluation is planned to demonstrate algorithms trained on this dataset perform robustly on other infrared datasets. Lastly, this camera setup uses a precise 940 nm bandpass filter on a monochrome color-filter array (CFA); however, it has recently become popular for security systems to employ a combined RGB-IR cameras for color day mode and monochrome night mode operation. This CFA inherently allows visible light to support the color pixels, introducing noise to the reflectance patterns. It would be pragmatic to also evaluate RGB-IR cameras to support new security systems. Both cases (external environment and camera CFA) would likely require a novel algorithm design to mitigate the noise factors. A series of future experiments are planned to incorporate these factors and augment the methodology.

## 7. Patents

This research has generated patent applications jointly filed between Ford Motor Company and the University of Michigan. If allowed, a patent number is provided; those that are still in process are identified by case ID.

1.  COUNTERFEIT IMAGE DETECTION (USPTO Case ID: 84238879US01). Convenience security facial-authentication using near-infrared camera specular reflectance. Person is first identified, then verified their compensated specular reflectance meets the liveliness-enrollment-similarity score.
2.  COUNTERFEIT IMAGE DETECTION (USPTO Case ID: 84227552US01). Secure facial-authentication capable of detecting complex 3D masks via co-registered CMOS and thermal cameras. CMOS camera is used to detect and identify the face; liveliness is determined using thermal analysis. System is secure with very efficient liveliness analysis.
3.  MATERIAL SPECTROSCOPY (USPTO Case ID: 84279449US01). Material source-identification using combined RGB-IR spectroscopy analysis. RGB provides material color context for near-infrared material spectroscopy. This provides a naive Anti-Spoofing approach (versus specular reflectance verification against enrollment).
4.  MATERIAL SPECTROSCOPY (USPTO Case ID: 84279422US01). Facial optical-tethering methods for material spectroscopy liveliness-analysis. Facial distance and orientation are determined using deterministic key-points or deep-learning.

5. MATERIAL SPECTROSCOPY (USPTO Case ID: 84279413US01). Facial environment-compensation methods for material spectroscopy liveliness analysis. Sequenced light toggling is used to detect the face with an illuminated frame and de-noise the background using non-illuminated frame analysis.
6. MATERIAL SPECTROSCOPY (USPTO Case ID: 84279409US01). Facial segmentation methods for material spectroscopy liveliness-analysis. In particular, emphasis is placed upon segmenting "skin" pixels either using deterministic key-points or semantically using deep-learning.

**Author Contributions:** Conceptualization, A.H. and J.D.; methodology, A.H. and H.M.; software, A.H.; validation, A.H. and J.D.; formal analysis, A.H.; investigation, H.M.; resources, J.D.; data curation, A.H.; writing—original draft preparation, A.H.; writing—review and editing, H.M.; visualization, A.H.; supervision, H.M. and J.D.; project administration, J.D.; funding acquisition, J.D. All authors have read and agreed to the published version of the manuscript.

**Funding:** The authors would like to give special thanks to Ford Motor Company for funding this research via University Alliance Grant, Biometric Forensics.

**Institutional Review Board Statement:** The University of Michigan IBR approved using facial images for face-recognition research.

**Informed Consent Statement:** Informed consent was obtained from all subjects involved in the study. All facial images are anonymized and stored on local hard drives.

**Data Availability Statement:** Full dataset is not available due to IBR requiring facial-images to be stored locally. Images containing just the authors can be provided upon request.

**Acknowledgments:** The authors greatly appreciate the participation of all who volunteered in the liveliness dataset collection. Ford engineers Justin Miller, Ian Parker, and Jun Lee are specifically thanked for their contribution in data collection and evaluation. Furthermore, the authors thank the authors of the Central Difference Convolutional Network, Learning Generalized Spoof Cues Network and Dual-Branch Meta-Learning Network for making their code available for research purposes. Their contributions are greatly appreciated, where all analysis in this paper is intending only to validate the proposed methodology.

**Conflicts of Interest:** Ali Hassani is both a graduate student at the University of Michigan-Dearborn and an employee of Ford Motor Company. This conflict of interest is reviewed by having independent principal investigators at both the university and Ford validate the deliverable.

## Abbreviations

The following abbreviations are used in this manuscript:

| | |
|---|---|
| FR | Face recognition |
| NIST | National Institute of Standards and Technologies |
| PAD | Presentation–attack–detection |

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
