# Peer review of "Monocular Facial Presentation–Attack–Detection: Classifying Near-Infrared Reflectance Patterns"

_applsci, doi:10.3390/app13031987_

Round 1

Reviewer 1 Report

This manuscript presents a monocular facial presentation-attack detection and a dataset for evaluation.

My comments are as follows:

1. The authors seem to assume Lambertian reflectance in their method (Section 3.1). However, the face does not obey pure Lambertian assumptions (such as eyes, beard, etc.), neither nor the spoofs mask and screen. 

2. Experiments are conducted on several existing classification methods, such as LBP, RFS, and deep learning models. In fact, this manuscript is more like a technical report but lacks contribution and novelty. It would be nice to see some new methods here.

3. Also, the settings of the used classification models are unclear, for example, how many samples are used in fine-turn? What do you mean fine-tune on the evaluation dataset? Do you mean you train and test the models on the same samples?!

4. In fact, some methods can decompose the surface shape and reflectance from a single view image (shape from shading) or images (photometric stereo). In this way, the authors can use both the 3D information (surface normal) and reflectance property for classification, without the use of 3D sensors. I suggest the authors add this part at least as a brief introduction in related work, with some reference:

[1] Woodham R J. Photometric method for determining surface orientation from multiple images[J]. Optical engineering, 1980, 19(1): 139-144.

[2] Ju Y, Shi B, Jian M, et al. Normattention-psn: A high-frequency region enhanced photometric stereo network with normalized attention[J]. International Journal of Computer Vision, 2022, 130(12): 3014-3034.

[3] Ramachandran V S. Perception of shape from shading[J]. Nature, 1988, 331(6152): 163-166.

Reviewer 2 Report

1. I think they should more contribution.

2. Add more reference recents.

Reviewer 3 Report

This manuscript deals with anti-spoofing using near-infrared images. It is an interesting work, but there are several unclear parts as below:

1. This manuscript generates a new dataset for anti spoofing as in Figure 4 using RGB and NIR images. It is unclear that why the section 3 is necessary. It is a kind of mathematical modeling and there is no experimental validation that this mathematical modeling is correct. Also, I cannot understand why model A, B and C are separately proposed. Since it seems to use RGB and NIR images directly for various methods as in section 4.2, the contents in section 3 make the reader confusing. 

2. Most algorithms considered in Section 4.2 are somewhat out-of-dated. Instead of LBP or DCT, it would be better to consider a new deep learning based algorithm to have better results on the target dataset. Also, it is not clear what is the actual input of deep learning networks. 

3. Based on previous comments, it is not clear that what is the contribution of this manuscript. If it is a new dataset generation, it would be better to make the dataset public. If the authors want to show the contribution of mathematical parts in section 3, experimental validations should be essential. 

Reviewer 4 Report

The paper presents an analysis of the Surface-reflectance in faces and spoof faces by using near-infrared light and a texture classifier in a monocular, single frame fashion. For this, the geometry and reflectivity of the surface of the face that contains convex and concave regions is considered.

Bibiographical references are limited and have not been reviewed as included in the paper, so there are doubts about the contributions of the paper.

Specific comments.

1. There are several writing errors. The wording must be reviewed and corrected. For example in Page 3 line 76 it is written “This is hypothesis is validated….”. The first “is” is misspelled.

2. Punctuation marks must be used at the end of each equation, as the case may be, either “.” either ",".

3. Page 6, line 237.- “…this research recommends 940 nm infrared because it has minimal contribution from solar radiance [38] or household LED light sources [39].” Despite what the authors comment in relation to the noise from solar radiance or LED lights, it cannot be neglected since the contribution in real applications complicate the face recognition process. The authors must include a sensitivity analysis of the concavity and convexity models on the facial surfaces analyzed under different levels of noise caused by solar radiance or LED light interference.

4. In the case of spoof faces, how do the different materials used in spoof masks affect if they are made of cotton cloth, paper, nylon or plastic, among other materials?

5. Author references are quite limited. There are quite a few previous works that have dealt with the topic of the paper related to face recognition analysis using spectroscopic and near infrared methods for different surface variations due to concavity and convexity. Some recommended articles to be analyzed are:

-          Gary A. Atkinson, Mark F. Hansen, Melvyn L. Smith, Lyndon N. Smith,

A efficient and practical 3D face scanner using near infrared and visible photometric stereo, Procedia Computer Science,Volume 2,2010,Pages 11-19

-          Mark F. Hansen, Gary A. Atkinson, Lyndon N. Smith, Melvyn L. Smith, 3D face reconstructions from photometric stereo using near infrared and visible light,

Computer Vision and Image Understanding, Volume 114, Issue 8, 2010,

Pages 942-951

-          LI, Stan Z., et al. Illumination invariant face recognition using near-infrared images.

IEEE Transactions on pattern analysis and machine intelligence, 2007, vol. 29, no 4, p. 627-639.

-          Peng, M.; Wang, C.; Chen, T.; Liu, G. NIRFaceNet: A Convolutional Neural

Network for Near-Infrared Face Identification. Information 2016, 7, 61.

https://doi.org/10.3390/info7040061

-          Tu, H.; Duoji, G.; Zhao, Q.; Wu, S. Improved Single Sample Per Person Face Recognition via Enriching Intra-Variation and Invariant Features. Appl. Sci. 2020, 10, 601.

https://doi.org/10.3390/app10020601

-          A near-infrared image based face recognition system, Li et al.

Conference Paper · May 2006 DOI: 10.1109/FGR.2006.13 · Source: IEEE Xplored

-          Recognizing an individual face: 3D shape contributes earlier than 2D surface reflectance information S Caharel, F Jiang, V Blanz, B Rossion - Neuroimage, 2009 - Elsevier

-          Richard Russell, Irving Biederman, Marissa Nederhouser, Pawan Sinha,

The utility of surface reflectance for the recognition of upright and inverted faces,

Vision Research, Volume 47, Issue 2,2007,Pages 157-165

-          Real-world face recognition: The importance of surface reflectance properties

R Russell, P Sinha - Perception, 2007

-          Neural correlates of shape and surface reflectance information in individual faces

F Jiang, L Dricot, V Blanz, R Goebel, B Rossion - Neuroscience, 2009 - Elsevier

-          Holistic processing of shape cues in face identification: Evidence from face inversion, composite faces, and acquired prosopagnosia F Jiang, V Blanz, B Rossion - Visual Cognition, 2011 - Taylor & Francis

- Effects of caricaturing in shape or color on familiarity decisions for familiar and unfamiliar faces ML Itz, SR Schweinberger, JM Kaufmann - PLoS One, 2016 - journals.plos.org

-          Surface curvature from kinetic depth can affect lightness. LM Peterson, DJ Kersten… - Journal of Experimental …, 2018 - psycnet.apa.org

-The inversion, part-whole, and composite effects reflect distinct perceptual mechanisms with varied relationships to face recognition. C Rezlescu, T Susilo, JB Wilmer… - Journal of Experimental …, 2017 - psycnet.apa.org

-          Contributions of shape and reflectance information to social judgments from faces

DW Oh, R Dotsch, A Todorov - Vision research, 2019 - Elsevier

-          Neural correlates of facilitations in face learning by selective caricaturing of facial shape or reflectance ML Itz, SR Schweinberger, C Schulz, JM Kaufmann - NeuroImage, 2014 - Elsevier

Round 2

Reviewer 1 Report

Can be accepted with current version

Reviewer 3 Report

All of my concerns are considered well in the revised manuscript. I recommend this manuscript for publishing in this journal.

Reviewer 4 Report

The authors have taken into account the comments related to the writing and drafting of the paper. Likewise, they have included recent bibliographical references and attended to the comments in relation to this point.